# Structure and mechanism of the broad spectrum CRISPR-associated ring nuclease Crn4

Haotian Chi [1,2], Ville Hoikkala[1,2], Stephen McMahon [1], Shirley Graham [1], Tracey Gloster [1] ✉ & Malcolm F. White [1] ✉

Type III CRISPR systems detect the presence of RNA from mobile genetic elements (MGE) in prokaryotes, providing antiviral immunity. On activation, the catalytic Cas10 subunit conjugates ATP to form cyclic oligoadenylate (cOA) signalling molecules that activate ancillary effectors, providing an immune response. Cellular ring nucleases degrade cOA to reset the system. Here, we describe the structure and mechanism of a new family of ring nucleases, Crn4, associated with type III-D CRISPR systems. The crystal structure of Crn4 reveals a small homodimeric protein with a fold unrelated to any known ring nuclease or, indeed, any known protein structure. Crn4 degrades a wide range of cOA species to linear oligoadenylates in vitro and ameliorates type III CRISPR immunity in vivo. Phage and plasmids also encode Crn4 orthologues that may function as anti-CRISPRs. These observations expand our understanding of ring nucleases and reveal a new protein fold for cyclic nucleotide recognition.

CRISPR-Cas systems provide adaptive immunity against invading mobile genetic elements (MGE) in prokaryotes[1]. Type III CRISPR-Cas systems comprise a ribonucleoprotein effector complex programmed with CRISPR RNA (crRNA) to detect invading RNA from viruses and other MGE[2]. Target RNA binding activates the catalytic Cas10 subunit, which in approximately 90% of cases harbours a specialised polymerase domain for the synthesis of cyclic oligoadenylate (cOA) molecules[3–6]. cOA acts as a second messenger, signalling viral infection in the cell by binding and activating a wide range of ancillary defence proteins (reviewed in ref. 7). Cleavage of bound target RNA deactivates the Cas10 cyclase activity[8] and extant cOA molecules are degraded by enzymes known as ring nucleases[9].

The Csm6/Csx1 family of type III CRISPR ancillary effectors use a dimeric CARF domain for cOA binding, resulting in allosteric activation of their HEPN-family ribonuclease domains[5,6]. Some of these effectors also degrade $cA_4$ or $cA_6$ in their CARF domains, providing a mechanism for auto-deactivation of the CRISPR immune response[10–14]. The effectors Cami1 and CalpL, which detect $cA_4$ via CARF and SAVED domains, respectively, also possess intrinsic ring nuclease activity[15–17].

The first extrinsic ring nuclease to be described, Crn1, was purified from *Saccharolobus solfataricus*, revealing a small protein with a dimeric CARF domain for cOA recognition. Crn1 slowly degrades cyclic tetra-adenylate ($cA_4$) to linear di-adenylate products[18]. A second family of extrinsic ring nucleases, Crn2, use a dimeric domain unrelated to CARF proteins to bind and degrade $cA_4$. Crn2 is found both in CRISPR defence operons[19–21] and in viral genomes, where it functions as an anti-CRISPR (Acr) by rapidly degrading $cA_4$ to neutralise cellular defences[21]. A third ring nuclease family, Crn3, uses a fold related to CARF domains to sandwich $cA_4$ in protein tetramers, degrading the signalling molecule in a metal-dependent reaction to linear di-adenylates[22,23]. Finally, a group of related proteins (Csx15, Csx16 and Csx20) were predicted to act as ring nucleases[24]. Recently, we demonstrated that Csx16 and Csx20 are $cA_4$-specific ring nucleases, and showed that the majority of $cA_4$- and $cA_6$-signalling type III CRISPR systems include a means to degrade their activator[25]. This has important implications for cellular outcomes on viral infection, as it provides a means to avoid programmed cell death or growth arrest[25,26].

We previously reported a systematic bioinformatic analysis of type III CRISPR systems, which resulted in the identification of a new

[1]School of Biology, University of St Andrews, BMS building, North Haugh, St Andrews, Fife, UK. [2]These authors contributed equally: Haotian Chi, Ville Hoikkala. ✉e-mail: tmg@st-andrews.ac.uk; mfw2@st-andrews.ac.uk

effector protein, Csm6-2, which is activated by cyclic hexa-adenylate ($cA_6$) and cleaves RNA non-specifically[3]. Csm6-2 is typically found associated with a CorA-family trans-membrane effector, and in some instances, the two proteins are fused. The CRISPR-associated CorA effector from *Bacteroides fragilis* was previously shown to be activated by SAM-AMP—a signalling molecule derived from conjugation of ATP and S-adenosyl methionine[27], but systems encoding both Csm6-2 and CorA may signal via $cA_6$[3].

Here, we show that these CRISPR loci include a small open reading frame encoding a hypothetical protein, hereafter named Crn4 (CRISPR ring nuclease 4), which degrades $cA_3$, $cA_4$ and $cA_6$ to a range of linear products. The crystal structure of Crn4 reveals a novel, dimeric fold with conserved histidine and arginine residues that are essential for catalysis. Crn4 can partially relieve CRISPR-mediated immunity in a plasmid challenge assay, consistent with a function as a cellular ring nuclease, and is encoded in some phage and plasmid genomes where it likely functions as an Acr.

## Results

### Identification of the CRISPR associated protein Crn4
Csm6-2, a fused Csm6 orthologue that is activated by $cA_6$ and cleaves RNA, was recently identified from a bioinformatic analysis of type III CRISPR loci[3]. A systematic scan of the regions neighbouring Csm6-2 revealed a small open reading frame (ORF) encoding a hypothetical protein of unknown function in 12 CRISPR type III-D loci. In 5 of the 12 instances, a CorA membrane effector was also present (Fig. 1a). In two cases, the ORF was fused with Csm6-2, and in one instance it was fused with the effector TIR-SAVED (Fig. 1b)[3,28]. A recent bioinformatic study identified this ORF, named Unk01, as a potential ring nuclease[24].

Using hidden Markov model (HMM) profiles based on these 12 homologues, we identified 10 additional homologues in our dataset, which were associated with the $cA_4$-activated effector Can2, and two homologues with no recognisable effector in the CRISPR locus. Phylogenetic analysis divided the 22 homologues into two distinct groups: group "a" associated with Csm6-2/CorA, and group "b" associated with Can2[29,30] or lacking a recognizable effector (Fig. 1a). The small size of the predicted proteins, combined with their consistent association with $cA_4$ and $cA_6$ activated effectors (either through adjacency or fusion) and prevalence on MGEs supported the prediction that Unk01 was a ring nuclease. Following established nomenclature, we provisionally designated this family of proteins CRISPR ring nuclease 4 (Crn4). The family was sub-divided into Crn4a and Crn4b, distinguishable as Crn4b members are smaller, lacking a ~ 10 amino acid region present in Crn4a (Supplementary Fig. 1). Crn4b is also observed in the genomes of phages and plasmids. Of the 28,137 phage genomes in the Millard database[31], 11 contained a Crn4 homologue, while this was true for 10 out of 59,895 plasmid sequences in the database PLSDB[32]. Some of the phage and plasmid orthologues have an 60–80 amino acid N-terminal extension of unknown function (Supplementary Fig. 2). All members of the family have conserved residues including T13, H15 and R112 (*A. procaprae* Crn4a numbering), which are implicated in cOA binding and catalysis (further details below).

### Crn4a is a ring nuclease with broad specificity
We chose candidate Crn4a protein WP_136192672, found in the *Actinomyces procaprae* type III CRISPR locus adjacent to the previously characterised Csm6-2 protein[3], for further study. To explore its function, we constructed a synthetic gene encoding the protein for expression in *Escherichia coli* and purified it by immobilised metal affinity and size exclusion chromatography (Supplementary Fig. 3). Crn4a was tested for ring nuclease activity by incubating the purified protein with synthetic cOA species under multiple turnover (substrate excess) conditions in EDTA, followed by HPLC analysis (Fig. 2a). Crn4a cleaved all three cOA species tested, $cA_6$, $cA_4$ and $cA_3$, generating linear products. $cA_6$ was degraded with a rate constant of

$0.024 \pm 0.007$ $min^{-1}$ (Fig. 2b and Supplementary Fig. 4), $cA_3$ slightly faster ($0.083 \pm 0.02$ $min^{-1}$) and $cA_4$ was degraded much more rapidly ($0.98 \pm 0.1$ $min^{-1}$). $cA_4$ and $cA_6$ were cleaved on both sides of the ring, yielding linear 2′,3′-cyclic phosphate $cA_2 > P$ and $cA_3 > P$ products, respectively (Fig. 2a and Supplementary Fig. 4). This suggests concerted or sequential cleavage of cOA on opposite sides of the molecules, reminiscent of the viral ring nuclease AcrIII-1[21]. $cA_3$, which unlike $cA_4$ and $cA_6$ lacks an axis of twofold symmetry, was degraded to $A_2 > P$ and $A_1 > P$, consistent with asymmetric cleavage (Fig. 2a and Supplementary Fig. 4). The ability to cleave multiple cOA species was not observed for Crn1-3[18,21,22]. Thus, in contrast to all other CRISPR associated ring nucleases tested to date, Crn4a has broad specificity for cOA species and is the first confirmed CRISPR-associated $cA_3$ ring nuclease.

We also cloned and expressed the gene encoding a representative of the Crn4b family (MBK8772583 from *Hyphomicrobiales bacterium*) (Supplementary Figs. 1 and 2). Crn4b degraded $cA_4$ and $cA_3$ but was inactive against $cA_6$ (Fig. 3a). This appears consistent with the association of Crn4b with the Can2 effector, which is activated by $cA_4$[29,30,33]. Although Crn4b produces the same final $cA_4$ cleavage products as Crn4a, intermediate linear products ($A_3 > P$ and $A_4 > P$) were observed at early reaction time points with $cA_3$ and $cA_4$, respectively (Fig. 3a and Supplementary Fig. 5), consistent with a sequential cleavage mechanism.

The close association of Crn4a with the Csm6-2 effector prompted us to test the ring nuclease activity of Csm6-2, as all Csm6 proteins studied to date possess intrinsic $cA_6$ ring nuclease activities in the CARF domain of the protein[10,11,13]. Assays revealed $cA_6$ degradation activity for Csm6-2 (Fig. 3b). The rate of cleavage of $cA_6$, $0.026 \pm 0.01$ $min^{-1}$ (Supplementary Fig. 6), was equivalent to that observed for Crn4a. However, neither $cA_3$ nor $cA_4$ were degraded by Csm6-2. This activity could reside in the CARF domain, HEPN domain or a combination of the two.

Given the broad specificity of Crn4a, we also tested its ability to degrade noncanonical substrates (Supplementary Fig. 7). Perhaps unsurprisingly, $cA_5$, which is not known to be an activator of any CRISPR effector, was degraded efficiently by Crn4a. The CBASS signalling molecule cAAG was not a substrate, suggesting a strict specificity for adenine bases, and a high concentration of Crn4 cleaved a 12 nt polyA RNA substrate three nucleotides from the 3′ end. This suggests RNA can fold into the active site and be cleaved by the enzyme. Previously, it was demonstrated that the CARF domain of the Csx1 ribonuclease can bind linear tetra-adenylate RNA tails, which activated the HEPN domain[34]. The physiological relevance of such observations is questionable, given the slow reaction kinetics.

### Structure and mechanism of Crn4
To characterise the structure of Crn4, we crystallised the wild-type Crn4a and Crn4b proteins in apo-form, along with the Crn4a H15A variant in the presence of $cA_6$. We collected X-ray diffraction data to 2.34, 1.09 and 1.44 Å resolution, respectively. The structures were solved using molecular replacement with the AlphaFold predicted structure as the model. The structures reveal a new protein fold, with no hits identified by the DALI server[35] with Z-scores higher than 2.6. The core structure comprises four (Crn4b; Fig. 4a) or five (Crn4a; Fig. 4b) pairs of beta strands, one parallel and the others anti-parallel, and one short alpha helix. Both proteins are dimeric, and strikingly, an extended β-hairpin, comprising one (Crn4b) or two (Crn4a) two-stranded anti-parallel beta sheets, from one monomer wraps around the neighbouring subunit, which is largely facilitated by hydrophobic interactions between the two molecules (Fig. 4a, b). The structures of apo Crn4a and Crn4b superimpose with an RMSD of 2.7 Å over 128 Cα residues. The only significant difference is the length of the hairpin loop, which is around 10 residues longer (at the N-terminus of loop) in Crn4a. The loop is unstructured in both proteins, and Crn4a and Crn4b

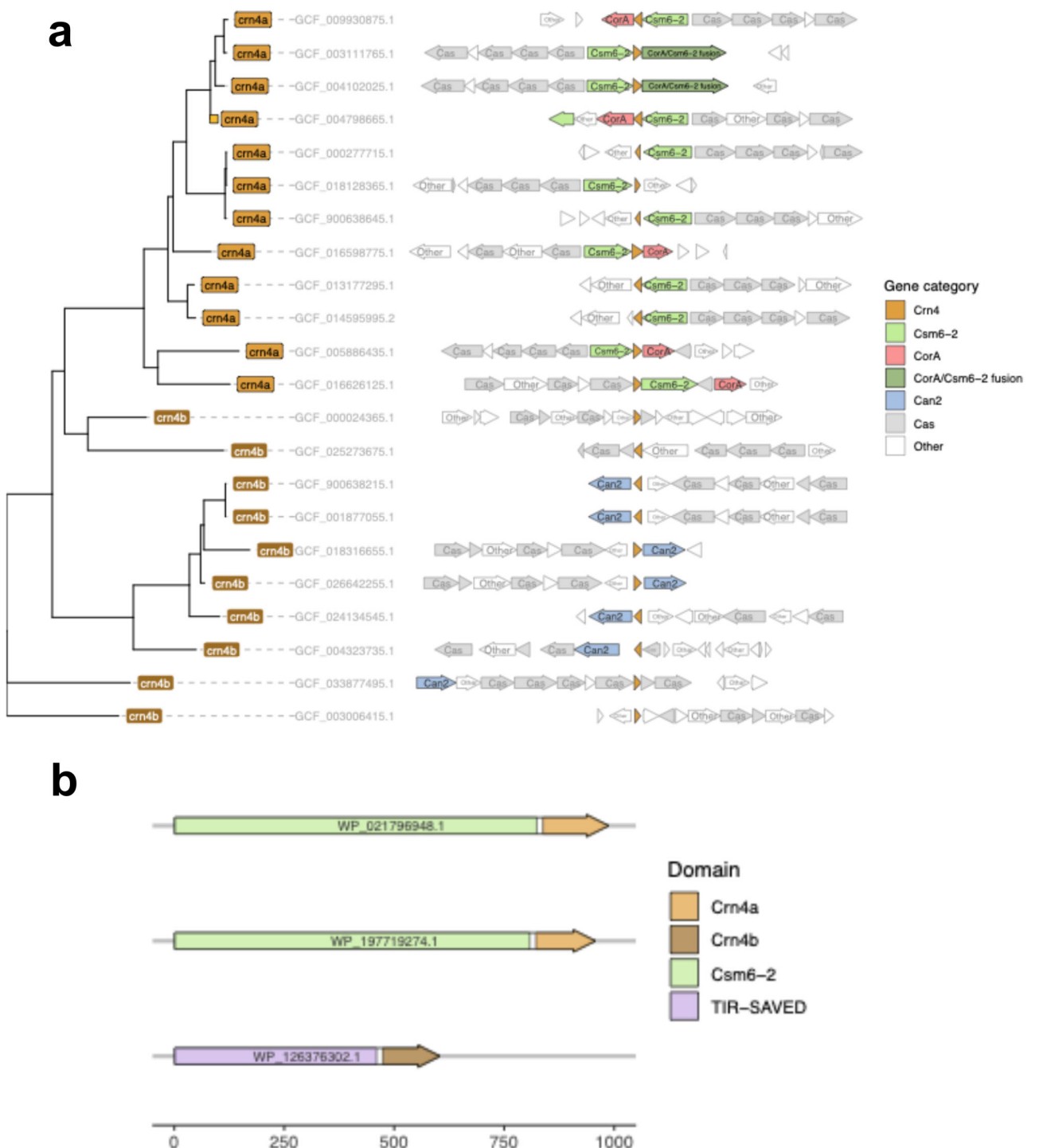

**Fig. 1 | Bioinformatic analysis of Crn4. a** A phylogenetic tree of Crn4 homologues found in type III CRISPR-Cas loci of complete prokaryotic genomes in NCBI. The subfamily of each Crn4, (**a** or **b**) is labelled on the end nodes followed by NCBI genomic accession numbers. The closest homologue (94% similarity) of the Crn4a tested experimentally in this study is marked by a small square. The genomic neighbourhoods of each Crn4, defined by CRISPR-Cas locus boundaries, is shown on the right, with key effectors highlighted as shown in the legend. **b** Three examples of Crn4 fused to the C-terminus of type III CRISPR-Cas effector proteins Csm6-2 and TIR-SAVED. Protein accession numbers are shown on the annotations.

have low sequence conservation in this region (Fig. 4c and Supplementary Fig. 8).

Upon structure solution, there was clear electron density visible in the maximum likelihood/σA weighted $F_{obs}$-$F_{calc}$ electron density map at 3σ corresponding to a molecule of cA$_6$ bound to Crn4a that originated from the crystals co-crystallised with the ligand (Supplementary Fig. 9A). The structure of Crn4a in complex with cA$_6$ shows the ligand binds at the dimer interface (ie. one molecule of cA$_6$ to two protein molecules) in a symmetrical compact arrangement, with two adenine bases oriented upwards, two downwards and two roughly axial to the plane of the cA$_6$ ring (Fig. 4d, e and Supplementary Fig. 9B). This differs from the conformation of cA$_6$ bound to *Streptococcus thermophilus* Csm6, where four of the six adenine bases were in the plane of the ring (Supplementary Fig. 9C)[36].

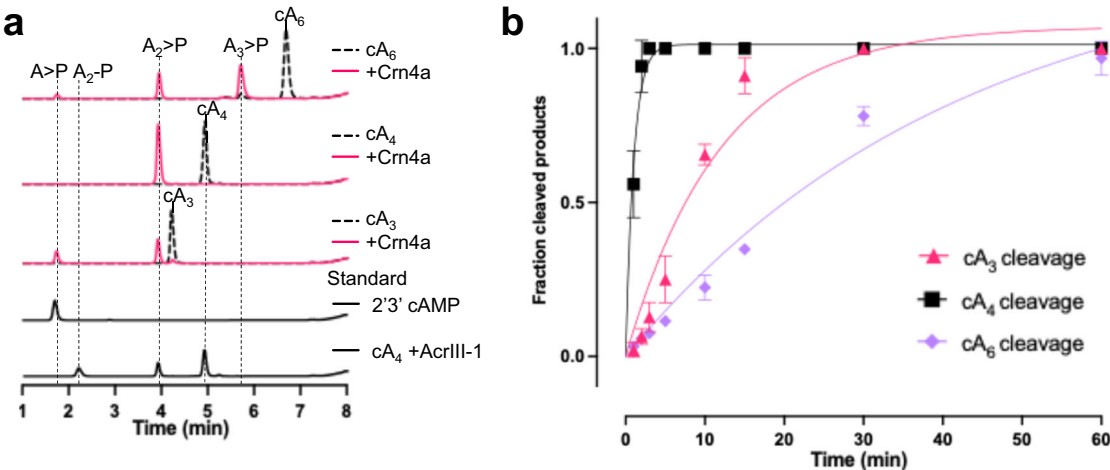

**Fig. 2 | Cleavage of cOA species by Crn4a. a** HPLC analysis of cleavage products of $cA_3$, $cA_4$ and $cA_6$ incubated with *A. procaprae* Crn4a. 2',3'-cAMP and a control reaction cleaving $cA_4$ with AcrIII-1 to generate $A_2 > P$ are shown as standards. **b** Kinetic analysis of ring nuclease activity of Crn4a against $cA_3$, $cA_4$ and $cA_6$.

Following HPLC, substrate and product peaks were quantified and data plotted as fraction cleaved against time. Data points represent the means of triplicate experiments and standard deviation is shown. Source data are provided as a Source Data file.

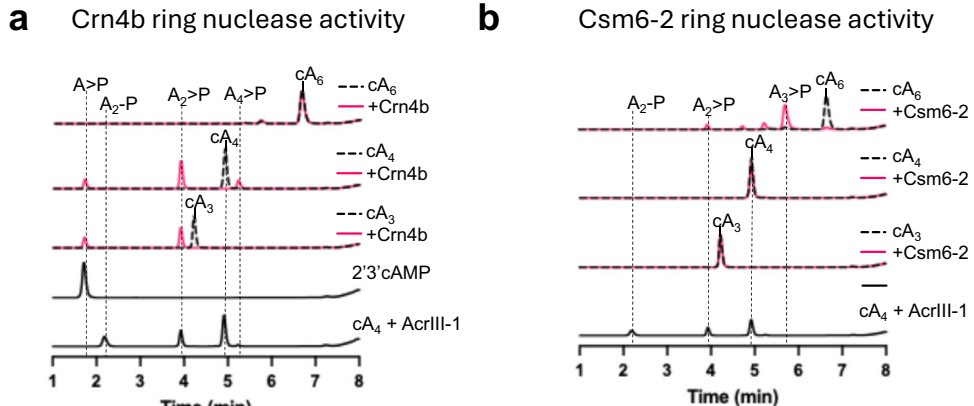

**Fig. 3 | Ring nuclease activity of Crn4b and Csm6-2. a** HPLC analysis of cleavage products of $cA_3$, $cA_4$ and $cA_6$ incubated with Crn4b. Controls are as for Fig. 2a. **b** *A. procaprae* Csm6-2 possesses ring nuclease activity against $cA_6$, but not $cA_3$ or $cA_4$. Source data are provided as a Source Data file.

In Crn4a, $cA_6$ hydrogen bonds with the main chain atoms of R92, E16, P77, A70, and side chain hydroxyl of S91, and a phosphate moiety forms an electrostatic interaction with R112 (Supplementary Fig. 9D). In addition, there is pi–pi stacking between an adenine of $cA_6$ and the guanidium group of R92. R92, E16, S91 and R112 are all structurally conserved in Crn4b, as are the main chain atoms of P77 and A70, but with alternative side chains. It is noteworthy that, given the size of $cA_6$, there are relatively few interactions formed with Crn4a, and the majority are with main chain atoms, suggesting the side chain functionality is not particularly important. Interestingly, R92 and R112 (or equivalent) side chains are found in a similar conformation in the two apo structures, but in a different orientation in the $cA_6$ complex with Crn4a, suggesting the residues are flexible and move upon ligand binding. Stabilisation of the negative charge of $cA_6$ is performed by two magnesium ions located in the centre of the $cA_6$. One of these interacts with two adjacent phosphate groups and two water molecules and the other with one phosphate group and three water molecules (Supplementary Fig. 9E).

The orientation of R112 in Crn4a, and its interaction with a phosphate group in $cA_6$, points to this being an important residue in phosphodiester bond cleavage (Fig. 4f). Alignment of the Crn4a H15A mutant (in complex with $cA_6$) and apo Crn4a suggests that His15 would

swing into a key position to participate in catalysis. A conserved threonine, T13, is positioned just behind the histidine. It is noteworthy that R112 and H15/T13 are located on different monomers in the dimer.

The inline angles formed by $O^{2'}$-P-$O^{5'}$ are between 100 and 110° around the ring, far from the angle of ~160° that is optimal for in-line attack by the 2'-OH nucleophile[21]. This may reflect the fact that $cA_6$ is not the optimal substrate for this enzyme; additionally, the H15A variant crystallised here may differ from the wild-type enzyme in this respect.

Comparison of the apo and $cA_6$-bound structures of a Crn4a monomer reveals that the globular part overlaps perfectly, but the extended β-hairpin bends upwards at an "elbow" when $cA_6$ is bound (Fig. 4g). In dimeric Crn4a, this movement translates to the protein encasing the cOA upon binding (Supplementary Movies 1 and 2). This conformational change means the loops of the β-hairpin come together to sit above the central cavity where $cA_6$ binds (Supplementary Fig. 9F). H15 and R112 are located on these loops, and move by 6 Å and 7 Å, respectively (Supplementary Fig. 9G), thus placing them in key positions to participate in cOA binding and/or cleavage. As H15 and R112 originate from different monomers in the Crn4a dimer, the precise loop movements from both are critical to form the active site, allowing catalysis to occur. Given the observation of relatively few

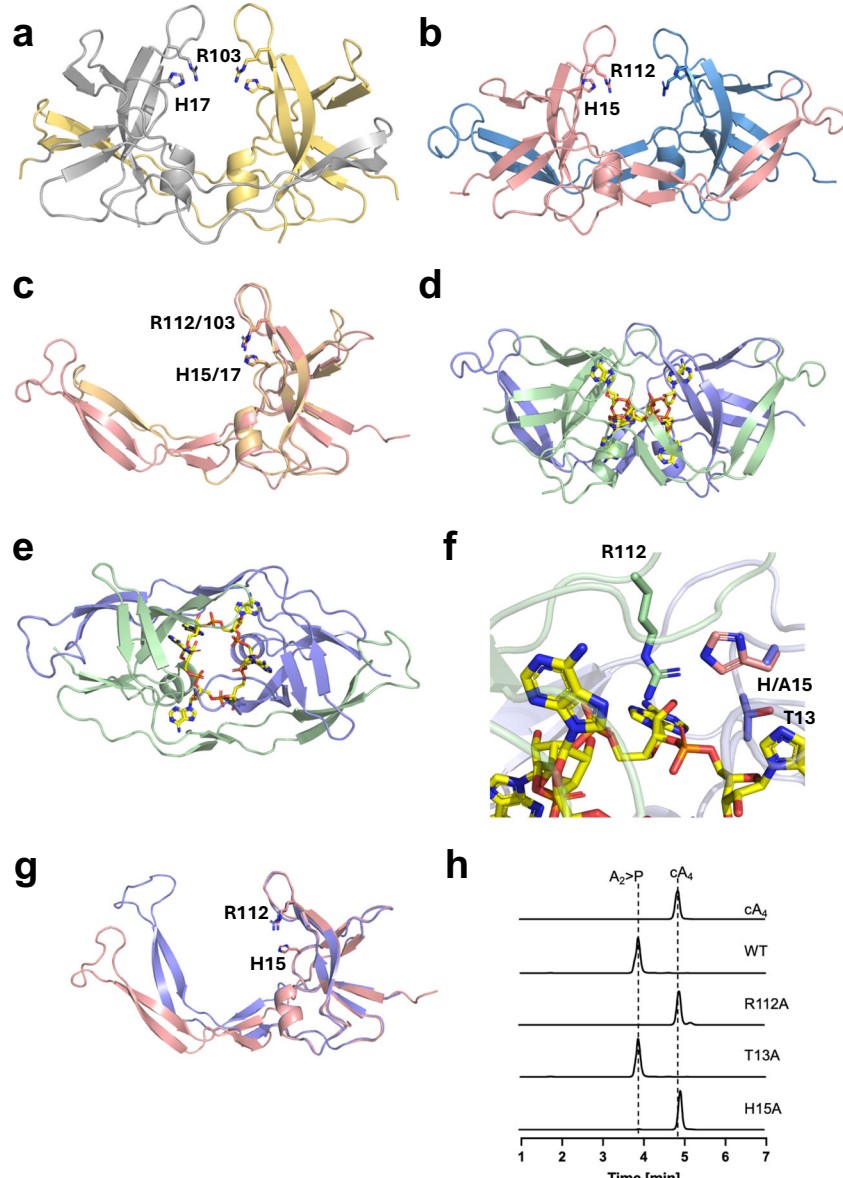

**Fig. 4 | Structure and mechanism of Crn4. a** Dimeric structure of Crn4b, with individual monomers coloured grey and gold. Each monomer has an extended β-hairpin arm wrapping around the other. The conserved residues H17 and R103 are positioned on loops above a central cavity. **b** Dimeric structure of Crn4a, with individual monomers coloured pink and blue. The conserved residues H15 and R1023 are positioned on loops above a central cavity. **c** Superimposition of a monomer of apo Crn4b (gold) and apo Crn4a (pink). Crn4a has a longer β-hairpin compared to Crn4b. The conserved residues H17 and R103 (Crnb4b) and H15 and R112 (Crn4a) are shown. **d** Dimeric structure of Crn4a in complex with cA₆, with monomers coloured mauve and green. cA₆ bound to Crn4a is shown in stick representation, with carbons coloured yellow. **e** Top-down view of the Crn4a dimer in complex with cA₆ (same colouring as **d**), highlighting how the extended β-hairpin wraps around the neighbouring monomer. **f** Active site of Crn4a in complex with cA₆, with monomers coloured green and mauve. R112 is suitably positioned to participate in cA₆ binding and catalysis. A15 (Crn4a mutant in ligand-bound structure) is shown in mauve and H15 from the apo Crn4a structure is shown in pink; there is space for the histidine to swing into place to play a role in catalysis. T13 sits behind H15. **g** Superimposition of a monomer of apo Crn4a (pink) and Crn4a bound to cA₆ (mauve), illustrating the movement of the β-hairpin upon binding of the ligand. Residues H15 and R112 are shown. **h** HPLC analysis of cleavage products of cA₄ incubated with wild type Crn4a and T13A, H15A and R112A variants. Mutagenesis of H15 and R112 abolished ring nuclease activity, while the T13A variant was still active.

interactions between cA₆ and Crn4a (Supplementary Fig. 4D), it is possible that the protein residues displayed in the "open" form of the dimer are more important for recognition and binding of cOA. Once the cOA is enclosed in the central cavity of the "closed" dimer there is less need for interactions to guide binding or specificity, and only those important for catalysis, or for correct positioning of catalytic residues, are important.

Despite our best efforts, we were unable to crystallise a ligand-bound complex with Crn4b, or indeed Crn4a in complex with cA₃ or cA₄. Given the considerable movement observed for Crn4a upon

binding of cA₆ and our suggestion that specificity may largely be driven by interactions with the 'open' form of Crn4, direct comparison of the factors influencing specificity for substrate for each protein was not possible. Given the broad substrate specificity of the enzyme, this remains an important topic for future study.

We constructed and purified variants T13A, H15A and R112A of Crn4a to test their role in catalysis. Mutation of H15 or R112 to alanine abolished cleavage of the preferred substrate, cA₄, after incubation at 37 °C for 60 min, while the T13A variant retained some activity (Fig. 4). Qualitatively similar data were obtained for cA₃ and cA₆ cleavage

(Supplementary Fig. 10). These data are consistent with a role for H15 as a general acid, protonating the oxyanion leaving group during phosphodiester bond cleavage, reminiscent of H47 in AcrIII-1[21]. R112 from the neighbouring monomer likely provides a key electrostatic interaction for the cOA species during binding and catalysis.

## Crn4 is functional in vivo

To explore a functional role of Crn4 in vivo, we reconstituted *A. procaprae* Crn4a into a well-developed recombinant type IIIA CRISPR system from *Mycobacterium tuberculosis* (MtbCsm), which produces a range of cOA species for effective activation of cOA-dependent effectors to provide plasmid immunity[37]. *E. coli* C43 cells expressing MtbCsm were transformed with a pRATDuet plasmid expressing an effector protein alone or along with Crn4a and carrying a tetracycline resistance gene (*tetR*). Fewer transformants were expected when the effector was activated by cOA generated by the MtbCsm system, which was programmed with a crRNA targeting a portion of *tetR* gene. As a control, we also tested MtbCsm programmed with a crRNA targeting the pUC plasmid, which is not activated to generate cOA in this experimental setup. Effectors tested in this programmed MtbCsm system were cA$_4$-activated Csx1 from *Thioalkalivibrio sulfidiphilus* (TsuCsx1)[27,38] and cA$_6$-activated effectors Csm6 from *M. tuberculosis* (MtbCsm6)[37] and *A. procaprae* Csm6-2[3].

In the absence of Crn4a, all three effectors were activated on transformation of the pRATDuet plasmid, providing effective resistance to plasmid transformation, reflected in reduced colony forming units (cfu) (Fig. 5 and Supplementary Fig. 11). The presence of Crn4a effectively alleviated plasmid immunity conferred by cOA-activated

TsuCsx1 and MtbCsm6 (Fig. 5), suggesting that Crn4a reduces the concentration of both cA$_4$ and cA$_6$ when expressed in cells. In contrast, no significant difference in colony counts was observed for the Csm6-2 effector in the presence or absence of Crn4a. This may be because Csm6-2 is itself a very active ring nuclease; alternatively, this could reflect the relative affinities of cA$_6$ binding by Crn4a and Csm6-2.

## Discussion

Ring nucleases, first described in 2018[18], are now understood to be an intrinsic aspect of type III CRISPR defence, where it seems that an ability to degrade the cOA second messengers generated by these systems is generally beneficial to the host[25]. Of the six types of specialised or stand-alone ring nucleases characterised to date, all are based on Rossmannoid domain architecture (e.g., CARF and SAVED domains)[25]. Furthermore, all of them are specific for degradation of cA$_4$, the most common cOA signalling species[3]. Meanwhile, the self-limiting effectors with intrinsic ring nuclease activity also use CARF or SAVED domains to degrade either cA$_6$ or cA$_4$[10–17], although the CalpL protein also has some activity against cA$_3$ in vitro[16]. Here, we have described a new family of ring nucleases, Crn4, with several notable characteristics. Firstly, Crn4 displays a novel (non-Rossmann) protein fold for cOA binding. Secondly, Crn4 has ring nuclease activity against all the cOA species known to signal in type III CRISPR defence.

The structure of Crn4 appears to be unique when compared to others in the Protein DataBank, expanding our understanding of the protein folds that can bind cyclic nucleotides. The chemistry of cOA cleavage, on the other hand, is highly reminiscent of previously characterised ring nucleases, such as AcrIII-1, with both enzymes

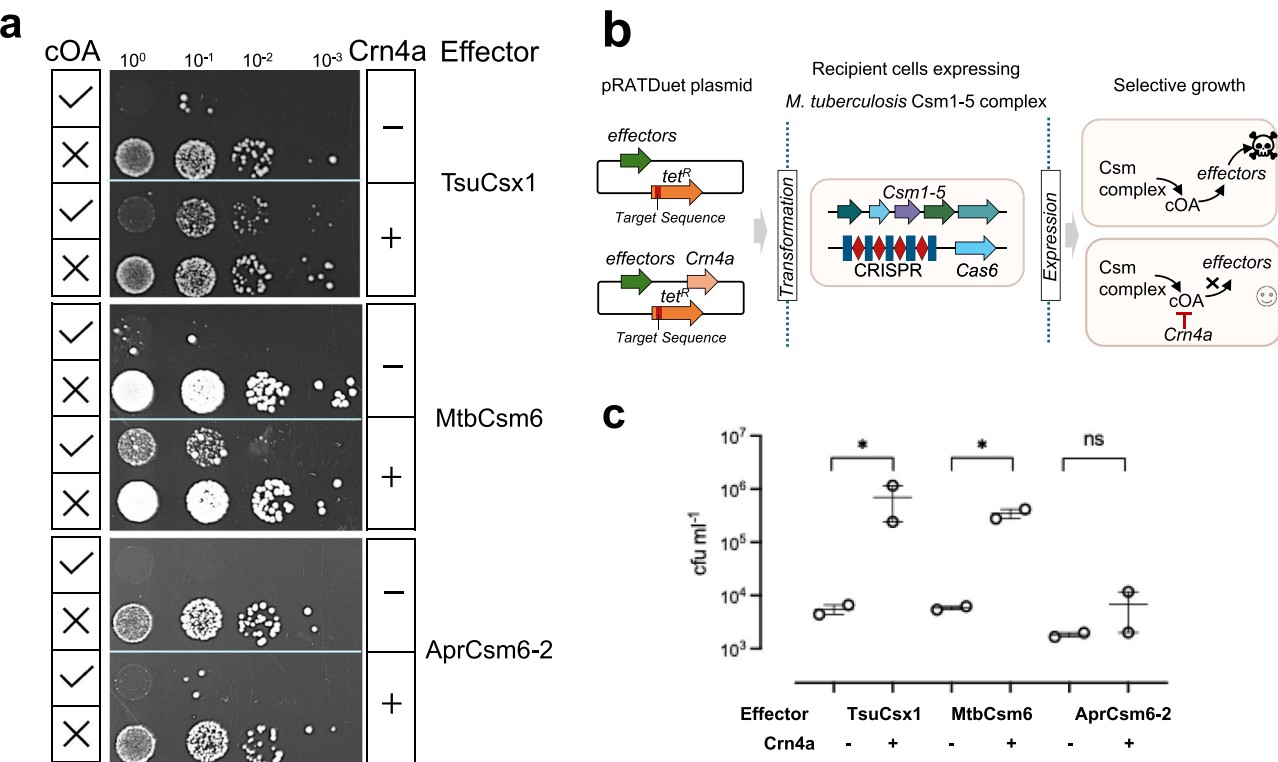

**Fig. 5 | Crn4a neutralizes CRISPR immunity in a plasmid challenge assay.**
**a** Plasmid challenge assay. The expression of Crn4a increased cfu in cells with cA$_4$-activated TsuCsx1 and cA$_6$-activated MtbCsm6, consistent with degradation of those cOA species. For the Csm6-2 effector, however, no increase in cfu was observed when Crn4a was co-expressed. cOA was synthesised in the MtbCsm system programmed with crRNA targeting the *tetR* gene, while no cOA production in the MtbCsm non-target (pUC) system was set as a negative control (represented by ticks and crosses, respectively). Representative plates of two biological replicates with two technical replicates each are shown. **b** Schematic of the experimental design. **c** Colony counts for transformants in the presence or absence of Crn4a in the context of activated MtbCsm target system. Data are two biological replicates with two technical replicates each and are presented as mean ± s.d. Statistical analysis was performed using ratio paired *t* test (one-tailed *P* values listed from left to right on the graph: *$P$ = 0.0399, *$P$ = 0.0217 and $^{ns}P$ = 0.25). Source data are provided as a Source Data file.

positioning an essential histidine as a likely general acid to facilitate catalysis[21]. The observation that $cA_3$ is degraded efficiently by Crn4 is particularly intriguing, given that the enzyme has a dimeric organisation with two identical active sites and twofold symmetry. While this is ideal for binding of twofold symmetric substrates such as $cA_4$ and $cA_6$, it poses a challenge for binding and cleavage of $cA_3$, which has three-fold symmetry. An understanding of $cA_3$ binding and cleavage at the molecular level requires a ligand-bound structure, but unfortunately, we were unsuccessful in obtaining this. In its absence, we can speculate that $cA_3$ engages part of the binding site and one of the two catalytic sites, leading to asymmetric cleavage. This is probably reflected in the cleavage of polyA RNA, which is cleaved 3 nucleotides in from the 3′-terminus, suggesting partial occupancy of the binding site and considerable substrate plasticity.

The detection of Crn4b homologues in phage and plasmid genomes implies a role for these enzymes as anti-CRISPR proteins, analogous to the $cA_4$-specific AcrIII-1[21]. The activity of Crn4b against both $cA_3$ and $cA_4$ suggests that both signalling molecules could be intercepted by these Acrs, although that remains to be confirmed. Viral nucleases targeting $cA_3$ could potentially function as anti-defence enzymes against both type III CRISPR and CBASS (cyclic nucleotide based anti-phage signalling systems), each of which utilise this signalling molecule[39–42]. To date, only the broad specificity anti-defence phosphodiesterase Acb1 has been shown capable of degrading $cA_3$[43].

We first identified Crn4 as a focus for investigation based on its close association with the $cA_6$-activated CRISPR effector Csm6-2[3]. As we have shown, Csm6-2 has intrinsic ring nuclease activity in its CARF domains and reaction rates in vitro are comparable to those for Crn4a, raising the question of why the latter protein is required? In fact, effectors with intrinsic ring nuclease activity are commonly found in type III CRISPR loci alongside dedicated ring nucleases[25], suggesting there is an advantage to the host organism to have such an arrangement. One possibility is that the relatively high levels of cOA species generated by an activated type III CRISPR system[8] necessitate a mechanism to remove these molecules when their function is no longer needed—one that does not rely on the activation, however transient, of a toxic effector protein.

In summary, these findings revealed a new class of cyclic nucleotide binding proteins, highlighting the wide distribution and diversity of ring nucleases in type III CRISPR defence.

## Methods

### Cloning
The g-blocks of the *crn4a* and *crn4b* genes were codon optimised for expression in *E. coli* and purchased from Integrated DNA Technologies. Synthetic genes were cloned into the vector pEhisV5TEV[8] between the *Nco*I and *BamH*I sites. The *E. coli* DH5a strain was used for cloning, and sequence integrity was confirmed by sequencing (Eurofins Genomics). Site-directed mutagenesis was performed using primers with the desired mutations. All synthetic genes and primers used in this study are listed in Supplementary Table 1. For the plasmid challenge assay, the g-block of the *crn4a* gene was cloned into the MCS-1 (Multiple Cloning Site-1) of pRATDuet[37] between the *Nde*I and *Xho*I sites, under the control of a T7 promoter. For the expression of two proteins, the synthetic gene encoding TsuCsx1[37], MtbCsm6[37] or Csm6-2C[3] was cloned into the MCS-2 (Multiple Cloning Site-2) of pRATDuet-Crn4 plasmid between the *Nco*I and *Hind*III sites, under the control of pBAD promoter.

### Protein expression and purification
*E. coli* C43 (DE3) cells were transformed with plasmids for protein expression. 1 L of Luria-Broth (LB) containing cells transformed with the plasmid of interest was grown at 37 °C to an $OD_{600}$ of 0.6–0.8. Crn4a or Crn4b expression was induced with 0.4 mM isopropyl b-D-1-thiogalactopyranoside for 4 h at 25 °C and 37 °C, respectively. Cell

pellets were harvested for purification by centrifugation at $4000 \times g$ (Beckman Coulter Avanti JXN-26; JLA8.1 rotor) at 4 °C for 15 min. Pellets were lysed by sonicating for 6 min with 1 min rest intervals on ice, before pelleting cell debris by ultracentrifugation at $160,000 \times g$ (Beckman Coulter Optima L-90K, 70 Ti rotor) at 4 °C for 30 min. Supernatants were subsequently loaded onto a 5 mL HisTrap FF column (GE Healthcare) equilibrated with lysis buffer. After washing away unbound proteins with 20 column volumes of lysis buffer, a linear gradient elution was conducted with pump A (lysis buffer) and B (elution buffer: 50 mM Tris-HCl pH 8.0, 0.5 M NaCl, 0.5 M imidazole, and 10% glycerol). Fractions containing recombinant protein were collected and concentrated for his-tag removal by overnight dialysis with TEV protease (1 mg per 10 mg protein) in size exclusion chromatography (SEC) buffer (20 mM Tris-HCl pH 8.0, 0.25 M NaCl, 1 mM DTT, and 10% glycerol). The TEV-cleaved protein was recovered by second immobilised metal affinity chromatography (IMAC) and further purified by SEC (S200 16/60 column, GE Healthcare) in SEC buffer under isocratic flow. The identity and purity of proteins were verified using SDS-PAGE and the pure protein was aliquoted and stored at −70 °C.

### Ring nuclease activity
To examine the ring nuclease activity, enzymes were incubated with 200-fold molar excess of cOA species in 20 mM Tris-HCl, pH 7.5, 250 mM NaCl and 1 mM EDTA at 37 °C for the time indicated. For calculation of rate constants of wild type Crn4a/b and Csm6-2, 500 nM enzyme was incubated with 100 $\mu$M $cA_3$, $cA_4$ or $cA_6$, at 37 °C for the time points 1, 2, 3, 5, 10, 15, 30 and 60 min. Reaction samples were quenched by mixing with two equivalent volumes of cold methanol and vortexed for 30 s, before centrifugation at 4 °C for 15 min. The supernatant was transferred to a new tube and vacuum dried, before resuspension in $H_2O$ for HPLC. Substrate and cleaved products were quantified from triplicate measurements and data were fitted using a one phase exponential model ($Y = Y_{max}*(1-e^{-k*x})$) using Prism (Version 10.2.2 (341)).

### HPLC
HPLC analysis was conducted on a UltiMate3000 UHPLC system (Thermo Fisher Scientific) equipped with an Accucore™ C18 column (Thermo Fisher Scientific 2.1 × 100 mm, particle size 2.6 $\mu$m) for the time-course samples and a C18 column (Kinetex EVO 2.1 × 50 mm, particle size 2.6 $\mu$m) for other analyses. The absorbance was monitored at 260 nm and the column temperature was set at 40 °C. Solvent A was 20 mM ammonium acetate, pH 8.5 and B was methanol. The flow rate was set at 0.4 ml/min for the Accucore™ C18 column and 0.3 ml/min for the EVO C18 column. Gradient elution followed the same procedure: 0–0.5 min, 1% B; 0.5–6 min, 1–15% B; 6–7 min, 100% B.

### Protein crystallisation
Crn4a H15A, at 11 mg mL$^{-1}$, was incubated at room temperature for 30 min with 1.2 molar excess of $cA_6$. Crn4a and Crn4b were used at concentrations of 13 and 15 mg mL$^{-1}$ respectively. Immediately prior to crystallisation proteins were centrifuged at $14000 \times g$. Sitting drop vapour diffusion experiments were set up at the nanoliter scale using commercially available crystallisation screens and incubated at 293 K. Optimised Crn4a H15A with $cA_6$ crystals were grown from 23.8% PEG 3350, 0.2 M magnesium chloride and 0.1 M Bis-Tris, pH 5.5. Apo Crn4a crystals were grown from 0.1 M sodium acetate, pH 4.5 and 22% v/v PEG Smear Broad (Molecular Dimensions). Crn4b crystals grew from 20% PEG 6000, 10% ethylene glycol and 0.1 M calcium chloride. Crystals were cryoprotected with 25% glycerol prior to harvesting and cryo-cooling in liquid nitrogen.

### X-ray data collection, structure solution, and refinement
X-ray data were collected at a wavelength of 0.9537 Å, 100 K, on beamline I04 at the Diamond Light Source, to 2.34 Å (apo Crn4a),

1.44 Å (Crn4a H15A with cA$_6$) and 1.09 Å (apo Crn4b) resolution. Data were automatically processed using Xia2[44]. The structures were solved by phasing the data using PhaserMR[45] in the CCP4 suite[46], using a model generated by AlphaFold 3[47], with initial B-factors modelled in Phenix[48]. Model refinement was achieved by iterative cycles of REFMAC5[49], where hydrogen atoms were added for refinement but not output to the pdb files, and manual model manipulation in COOT[50]. For the Crn4a H15A with cA$_6$ data, electron density for cA$_6$ was clearly visible in the maximum likelihood/σA weighted $F_{obs}$-$F_{calc}$ electron density map at 3σ. The coordinates for cA$_6$ were generated in Chem-Draw (Perkin Elmer) and the library was generated using Acedrg[51], before fitting the molecule in COOT. The quality of each structure was monitored throughout using Molprobity[52]. Data and refinement statistics are shown in Supplementary Table 2. The coordinates and raw data have been validated and deposited in the Protein Data Bank with deposition codes 9SMA for Crn4a, 9QS9 for Crn4a H15A in complex with cA$_6$, and 9R7B for Crn4b.

## Plasmid challenge assay

Plasmids from the programmed type III MtbCsm system have been described, including pCsm1-5 (containing Csm interference genes *cas10*, *csm3*, *csm4* and *csm5* from *M. tuberculosis* and *csm2* from *M. canettii*)[37], pCRISPR-TetR and pCRISPR-pUC (containing *M. tuberculosis cas6* and a CRISPR array targeting a tetracycline-resistance gene or a pUC multiple cloning site, respectively)[37] and pRATDuet-effectors (carrying tetracycline-resistance gene and *M. tuberculosis* (Mtb) *csm6* or *Thioalkalivibrio sulfidiphilus* (Tsu) *csx1*)[53].

*E. coli* C43 (DE3) cells carrying plasmids pCsm1-5 and pCRISPR-TetR or pCRISPR-pUC were transformed with 100 ng of pRATDuet derived plasmids containing different effectors with or without a Crn4 ring nuclease. After recovering at 37 °C for 2 h, a tenfold serial dilution of cells was applied onto LB agar supplemented with 100 µg/ml ampicillin and 50 µg/ml spectinomycin for determination of cell density of recipients, onto LB agar containing additional 12.5 µg/ml tetracycline for determination of transformation efficiency and LB agar with additional 0.2% (w/v) D-lactose and 0.2 % (w/v) L-arabinose to determine the plasmid immunity mediated by cOA-dependent effectors. Plates were incubated at 37 °C overnight, before taking images of plates. Technical duplicates of two biological replicates were performed. Colonies were counted to calculate colony forming units (cfu/ml) and analysed using an unpaired *t* test to compare the difference between systems in the presence or absence of Crn4 using Prism (Version 10.2.2).

## Bioinformatic analyses

To search for Crn4 homologues in prokaryotes, we downloaded the 38,742 complete prokaryotic genomes from NCBI on March 25th 2024. For phage genomes we used the Millard database of 28,114 genomes (May 2024 release)[31] and for plasmids the PLSDB database containing 59,895 sequences[32]. Based on our initial identification of two main clades for Crn4 homologues, separate HMM databases were created for Crn4a and Crn4b. These profiles were integrated into a previously published Snakemake pipeline developed for finding type III CRISPR-Cas loci, effectors and ring nucleases in prokaryotic and in phage genomes[3]. In short, the pipeline annotates type III CRISPR-Cas loci and searches them for ring nucleases based on HMM profiles and protein length limitations.

To create the phylogenetic tree (Fig. 1), prokaryotic Crn4a and Crn4b amino acid sequences were concatenated into one file, and aligned using Muscle[54]. A phylogenetic tree based on the alignment was created with FastTree[55] using -wag and -gamma arguments. Two Crn4 homologues (genomes GCF_002285795.1 and GCF_002863905.1) were removed from the dataset due to missing cyclase domains in the accompanying Cas10. The tree was plotted and annotated in RStudio v. XX with ggtree[56]. The accompanying genomic neighbourhood visualisation was created by mergingh information from several annotation files output by the pipeline (CCTyper, cATyper, RN_typer) into the original NCBI general features files (.gff). The enhanced annotations were visualised using ggtree's geom_facet wrapper for gggenes and centered around Crn4. Custom rules were used to bin the genes into the categories shown in Fig. 1. Crn4 fusions with known effector proteins were discovered by modifying the snakemake pipeline to search for Crn4 without protein length limitations.

## Reporting summary

Further information on research design is available in the Nature Portfolio Reporting Summary linked to this article.

## Data availability

Unless otherwise stated, all data supporting the results of this study can be found in the article, supplementary, and source data files. The protein structure coordinates and data have been deposited in the Protein Data Bank [www.pdb.org] with deposition codes 9SMA for apo Crn4a, 9QS9 for Crn4a H15A in complex with cA$_6$ and 9R7B for Crn4b. Source data are provided with this paper.

## Code availability

The code used for bioinformatic analyses has been uploaded to Zenodo [https://doi.org/10.5281/zenodo.17723392].

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

## Acknowledgements

Thanks to Dr Sabine Grüschow for helpful discussion. This work was supported by a European Research Council Advanced Grant (Grant REF 101018608 to M.F.W.).

## Author contributions

V.H. wrote the code and carried out the bioinformatic analysis. HC performed the cloning, expression, purification and biochemical characterisation with the assistance of S.G. S.M. performed the crystallisation, data collection and X-ray structure solution. T.G. and M.F.W. conceptualised the project and interpreted the data. All authors contributed to the drafting and revision of the manuscript.

## Competing interests

The authors declare no competing interests.
