## [Transparent Peer Review file · Nature Communications]

Structure and mechanism of the broad spectrum CRISPR-associated ring nuclease Crn4

Corresponding Author: Professor Malcolm White

Version 0:

Reviewer comments:

Reviewer #1

(Remarks to the Author)

In this manuscript, Chi et al. report the discovery and characterization of a new CRISPR ring nuclease Crn4. Through bioinformatic analysis, authors found two subclasses of Crn4, Crn4a and Crn4b that associated with the Csm6-2/CorA and Can2 effectors, respectively. They showed that both Crn4a and Crn4b can cleave the cOA molecules, with Crn4a targeting all cA3, cA4 and cA6 while Crn4b targeting only cA3 and cA4. These properties make Crn4 family of ring nucleases unique due to their broad specificity and targeting cA3. Authors presented two ultra-high-resolution crystal structures, that of Crn4b and that of Crn4a-H15A-cA6, revealing their unique fold and cA6-recognition mechanism. Three of the identified key interaction residues were mutated on Crn4a and the resulting variants were tested for cOA cleavage activities. Finally, authors performed a plasmid challenge assay to demonstrate the function of the Crn4 ring nuclease in cells. Overall, this manuscript is clearly written in a structured manner. The majority of experimental results support authors' conclusions and contribute to our understanding of the CRISPR associated ring nucleases. However, several concerns should be addressed prior to publication.

Major concerns:

- 1) Given the broad specificity of Crn4a, it may be of interests to know if it also cleaves linear polyA or even other cyclic nucleotides, perhaps at elevated concentrations;
- 2) The structural analysis did not provide a rationale for the difference in cOA cleavage specificity between Crn4a and Crn4b or between Crn4 and other ring nucleases;
- 3) It would be helpful to clarify, either by structural argument or additional experiments, if the b-hairpin conformation is an inherent difference between Crn4a and Crn4b or due to cA6-binding to Crn4a.
- 4) Along the same line, can new experiments support that the difference in b-hairpin contribute to or account for the observed difference in cOA specificity between Crn4a and Crn4b?
- 5) The 1.09 – 1.44 Å are likely among the highest resolutions of currently known cOA-interacting enzymes. However, authors did not take advantage of the high quality structures in revealing previously unknown features. Authors did not show a single electron density map, for instance, around any described structural elements (i.e, protein-protein or cA6 interfaces). Did authors add hydrogens during refinement and observed clear hydrogen bonds responsible for specificity? Are there metal ions observed, for instance, mediating binding of cA6? Are there any water molecules around the cleavage sites? Are the two scissile phosphate bonds in an in-line conformation (after modeling His15 in)?
- 6) Given the high resolutions, authors should list amino acid rotamer outlier statistics in Table S2. It would be interesting to see if there are any meaningful outliers observed at these resolutions.
- 7) LC-MS analysis was mentioned in the method but no experimental data were shown.

Minor comments

1. Figure 4 (especially panels A-D) needs proper labels.
2. The manuscript can benefit from a supplementary figure showing structural comparison among the Crn/Csm6/Csx1 proteins, especially their cOA-binding modes. The conserved residue among them should be compared.
3. Add secondary structure elements to the aligned sequences of Crn4 in Supplementary Figure 1. Highlight the b-hairpin mediating dimerization in both enzymes.

Reviewer #2

(Remarks to the Author)

Reviewer #3

(Remarks to the Author)

Reviewer's Comments

This manuscript by Chi et al. presents a comprehensive study on Crn4, a novel family of ring nucleases associated with type III-D CRISPR systems, combining bioinformatics, biochemistry, structural biology, and in vivo functional assays. The identification of Crn4 with a unique protein fold and broad cOA substrate specificity (including cA₃, cA₄, and cA₆) significantly advances our understanding of type III CRISPR signaling and regulation. The work is rigorous and impactful, but several key revisions are needed before it can be considered for publication in Nat Commun.

Major points:

1. The structure of Crn4a presents as a homodimeric assembly and shows symmetric binding for cA₆, with four adenine bases in the plane of the ring, and the remaining two axial to ring. This symmetric binding mode provides a structural basis for its recognition of cA₆ and cA₄, which both possess an axis of two-fold symmetry. Strikingly, cA₃, which lacks such two-fold symmetry, is degraded efficiently by Crn4a to A₂>P and A₁>P, indicating asymmetric cleavage. Therefore, the structure of cA₃-bound Crn4a would be quite valuable to directly visualize how this asymmetric cleavage is achieved. If obtaining this structure is not feasible, can the authors provide a detailed discussion to address this point?

2. Page 6, 2nd paragraph: the authors drew a conclusion that Csm6-2 cleaves cOA using its CARF domain rather than HEPN domain, based on the result that EDTA did not inhibit cOA cleavage. However, as previously noted (Du et al. 2024, 43, 304), EDTA does not effectively inhibit HEPN domain RNase activity of TtCsm6. I recommend using a Csm6-2 mutant lacking HEPN domain activity to specifically address which domain is responsible for its intrinsic ring nuclease activity on cA₆.

3. Page 7, 1st paragraph: In the structural analysis of cA₆ binding, the authors identify several residues (Arg92, Glu16, Ser91, Arg112, etc.) involved in cA₆ interaction. However, these residues are not indicated in the figures. Please explicitly label these key residues in Fig. 4C and 4D to visually link the text description to the structural context. Also please show the labeling for residues in Fig. 4A-B

4. To support the claim of specific cA₆ binding, please include a panel in Fig. 4 showing the 2F_o-Fc electron density map for the bound cA₆ ligand.

5. Please standardize to residue abbreviations (one-letter or three-letter) in all text, figure legends, and labels.

Version 1:

Reviewer comments:

Reviewer #1

(Remarks to the Author)

Authors added a new crystal structure of the apo Crn4a, allowing structural comparison. Authors also performed cleavage experiments to compare substrate specificities. Additional, all minor, comments are included below.

Did authors intend to use "surprisingly" instead of "unsurprisingly" in the sentence "Perhaps unsurprisingly, cA₅, which is not known to be an activator of any CRISPR effector, was degraded efficiently by Crn4a." ?

It is better to add the requirement for high-concentrations of polyA in the sentence "12 nt polyA RNA substrate was cleaved by Crn4a three nucleotides from the 3' end at an elevated concentration."

Authors could note that "The fact that Crn4a, which lacks a HEPN domain, can cleave polyA suggests its substrate plasticity."

Figure S7. Panel A, can authors clarify what the cleavage products of cA₃ and cA₅ likely are? They do not seem to be aligned on the chromatogram.

Reviewer #2

(Remarks to the Author)

Reviewer #3

(Remarks to the Author)

All of my concerns have been adequately addressed in the revised manuscript. I have no further comments and recommend acceptance for publication.

Response to REVIEWER COMMENTS

We thank all the reviewers for their expert and thoughtful comments. We address each of these point by point below. We feel that the resultant revised manuscript is greatly improved due to their input.

Along with all required textual changes we have:

- Obtained and described a new crystal structure of Crn4a in the apo form, allowing detailed analysis of the conformational changes on cA6 binding.
- Added further data showing that Crn4a can cleave cA5 and polyA RNA.
- Added further analysis of the structures as requested.

Reviewer #1 (Remarks to the Author):

In this manuscript, Chi et al. report the discovery and characterization of a new CRISPR ring nuclease Crn4. Through bioinformatic analysis, authors found two subclasses of Crn4, Crn4a and Crn4b that associated with the Csm6-2/CorA and Can2 effectors, respectively. They showed that both Crn4a and Crn4b can cleave the cOA molecules, with Crn4a targeting all cA3, cA4 and cA6 while Crn4b targeting only cA3 and cA4. These properties make Crn4 family of ring nucleases unique due to their broad specificity and targeting cA3. Authors presented two ultra-high-resolution crystal structures, that of Crn4b and that of Crn4a-H15A-cA6, revealing their unique fold and cA6-recognition mechanism. Three of the identified key interaction residues were mutated on Crn4a and the resulting variants were tested for cOA cleavage activities. Finally, authors performed a plasmid challenge assay to demonstrate the function of the Crn4 ring nuclease in cells. Overall, this manuscript is clearly written in a structured manner. The majority of experimental results support authors' conclusions and contribute to our understanding of the CRISPR associated ring nucleases. However, several concerns should be addressed prior to publication.

Major concerns:

1) Given the broad specificity of Crn4a, it may be of interests to know if it also cleaves linear polyA or even other cyclic nucleotides, perhaps at elevated concentrations;

As suggested, we tested Crn4a against linear polyA. At high concentrations, we observed cleavage of the substrate to generate a linear A3 product, suggesting that linear polyA can serve as a substrate mimic of cOA species. We also tested for cleavage of cAAG, but observed no cleavage, suggesting a strict preference for adenine bases. Finally, we tested for cleavage of cA5, which is synthesised by some type III CRISPR systems but which has no known role as an activator – this was cleaved efficiently by Crn4a. These new data are shown in Supplementary Figure 7 and described in the text:

Given the broad specificity of Crn4a, we also tested its ability to degrade noncanonical substrates (Supplementary Figure 7). Perhaps unsurprisingly, cA₅, which is not known to be an activator of any CRISPR effector, was degraded efficiently by Crn4a. The CBASS signalling molecule cAAG was not a substrate, suggesting a strict specificity for adenine bases, and a 12 nt polyA RNA substrate was cleaved by Crn4a three nucleotides from the 3' end. This suggests RNA can fold

into the active site and be cleaved by the enzyme. Previously, it was demonstrated that the CARF domain of the Csx1 ribonuclease can bind linear tetra-adenylate RNA tails which activated the HEPN domain³⁴. The physiological relevance of such observations is questionable, given the slow reaction kinetics.

2) The structural analysis did not provide a rationale for the difference in cOA cleavage specificity between Crn4a and Crn4b or between Crn4 and other ring nucleases;

A good understanding of the relationship between cOA specificity and structure for the two ring nucleases would require structures with cA3 or cA4 bound in each. Unfortunately, although we attempted to obtain such complexes for both Crn4a and Crn4b, the data always showed no ligand bound. In addition, the large conformational change observed upon binding of cA₆ suggests that it is residues in the 'open' form of the dimer that is likely to be important for defining specificity, rather than those seen in the closed form. This point is now discussed in the text:

Despite our best efforts, we were unable to crystallise obtain a ligand-bound complex with Crn4b, or indeed Crn4a in complex with cA₃ or cA₄. Given the considerable movement observed for Crn4a upon binding of cA₆, and our suggestion that specificity may largely be driven by interactions with the 'open' form of Crn4, direct comparison of the factors influencing specificity for substrate for each protein was not possible. Given the broad substrate specificity of the enzyme, this remains an important topic for future study.

3) It would be helpful to clarify, either by structural argument or additional experiments, if the b-hairpin conformation is an inherent difference between Crn4a and Crn4b or due to cA₆-binding to Crn4a.

4) Along the same line, can new experiments support that the difference in b-hairpin contribute to or account for the observed difference in cOA specificity between Crn4a and Crn4b?

To address comments 3 and 4, we solved the crystal structure of apo Crn4a (PDB 9SMA) allowing direct comparisons to be made with the ligand-bound structure. These are best viewed in Supplemental movies 1 and 2.

Addressing point 3: the beta-hairpin conformation is similar in apo Crn4a and Crn4b structures, but moves upon binding of cOA to Crn4a. We have added the following to the text:

Comparison of the apo and cA₆-bound structures of a Crn4a monomer reveals that the globular part overlaps perfectly, but the extended b-hairpin bends upwards at an "elbow" when cA₆ is bound (Figure 4G). In dimeric Crn4a, this movement translates to the protein encasing the cyclic oligoadenylate upon binding (Supplementary movies 1 and 2). This conformational change means the loops of the b-hairpins come together to sit above the central cavity where cA₆ binds (Supplementary Figure 9F).

Addressing point 4: both Crn4a and Crn4b cleave cA₃ and cA₄, so the only difference in specificity is Crn4a's ability to cleave cA₆. Superimposition of the apo Crn4a and Crn4b structures (see new Figure 4C) shows the catalytic residues superimpose perfectly, suggesting from a catalytic machinery viewpoint there is sufficient space for Crn4b to accommodate cA₆. Our current hypothesis is that the specificity is likely determined by the 'open' form of the dimer before the cOA becomes enclosed in the binding cavity. We have added text to this effect:

Given the observation of relatively few interactions between cA₆ and Crn4a (see above, Supplementary Figure 4D), it is possible that the protein residues displayed in the 'open' form of the dimer are more important for recognition and binding of cOA, and once the cOA has been enclosed in the central cavity of the 'closed' dimer there is less need for interactions to guide binding or specificity, and only those important for catalysis, or for correct positioning of catalytic residues, are required.

5) The 1.09 – 1.44 Å are likely among the highest resolutions of currently known cOA-interacting enzymes. However, authors did not take advantage of the high quality structures in revealing previously unknown features. Authors did not show a single electron density map, for instance, around any described structural elements (i.e., protein-protein or cA₆ interfaces). Did authors add hydrogens during refinement and observed clear hydrogen bonds responsible for specificity? Are there metal ions observed, for instance, mediating binding of cA₆? Are there any water molecules around the cleavage sites? Are the two scissile phosphate bonds in an in-line conformation (after modeling His15 in)?

We have added an electron density figure (Supplementary Figure 9A) showing the difference map for cA₆.

Hydrogen atoms were included in refinement throughout, but not written out to the PDB file. A sentence has been added to this effect in the Methods section:

Model refinement was achieved by iterative cycles of REFMAC5⁴⁹, where hydrogen atoms were added for refinement but not output to the pdb files, and manual model manipulation in COOT⁵⁰.

Hydrogen bond interactions are discussed in the text and are shown in new Supplementary Figures 9D and 9E. Relatively few interactions are seen between cA₆ and Crn4a, and often through main chain atoms, which led us to the hypothesis that the specificity is largely determined by the 'open' form of the dimer and once cA₆ is enclosed fewer interactions are required other than to orientate the catalytic residues. Additional text shown in point 4 above has been added to the manuscript to discuss this.

There are two magnesium ions coordinated to two of the phosphate groups in cA₆, which have been shown in new Supplementary Figure 9E. These magnesium ions interact with three water molecules, which have also been included in the figure.

Finally, for the question about in-line attack, we have measured all the in-line angles, which are in the range 100-100 °, far from the optimum for in-line attack. The following text has been added to the results for this:

The inline angles formed by O^{2'}-P-O^{5'} are between 100-110 ° around the ring, far from the angle of ~160 ° that is optimal for in-line attack by the 2'-OH nucleophile ²¹. This may reflect the fact that cA₆ is not the optimal substrate for this enzyme; additionally, the H15A variant crystallised here may differ from the wild-type enzyme in this respect.

6) Given the high resolutions, authors should list amino acid rotamer outlier statistics in Table S2. It would be interesting to see if there are any meaningful outliers observed at these resolutions.

Rotamer outlier statistics have been added to Table S2. There was a single rotamer outlier in the structure of Crn4a in complex with cA₆, but the rotamer was not deemed to be meaningful in the context of the article.

7) LC-MS analysis was mentioned in the method but no experimental data were shown.

The referee is correct, this method was not used in the manuscript and has been removed in the revision.

Minor comments

1. Figure 4 (especially panels A-D) needs proper labels.

Figure 4 (and associated supplementary figures) have been revised extensively, in response to this and other suggestions.

2. The manuscript can benefit from a supplementary figure showing structural comparison among the Crn/Csm6/Csx1 proteins, especially their cOA-binding modes. The conserved residue among them should be compared.

We have considered this suggestion, but as Crn4 is completely unrelated to the other known ring nucleases we feel this type of comparison might be better suited to a review paper.

3. Add secondary structure elements to the aligned sequences of Crn4 in Supplementary Figure 1. Highlight the b-hairpin mediating dimerization in both enzymes.

We have added a figure to the supplementary data showing these details, as requested (see Supplementary Figure 8).

Reviewer #2 (Remarks to the Author):

Reviewer #3 (Remarks to the Author):

Reviewer's Comments

This manuscript by Chi et al. presents a comprehensive study on Crn4, a novel family of ring nucleases associated with type III-D CRISPR systems, combining bioinformatics, biochemistry, structural biology, and in vivo functional assays. The identification of Crn4 with a unique protein fold and broad cOA substrate specificity (including cA₃, cA₄, and cA₆) significantly advances our understanding of type III CRISPR signaling and regulation. The work is rigorous and impactful, but several key revisions are needed before it can be considered for publication in Nat Commun.

Major points:

1. The structure of Crn4a presents as a homodimeric assembly and shows symmetric binding for cA₆, with four adenine bases in the plane of the ring, and the remaining two axial to ring. This symmetric binding mode provides a structural basis for its recognition of cA₆ and cA₄, which both possess an axis of two-fold symmetry. Strikingly, cA₃, which lacks such two-fold symmetry, is degraded efficiently by Crn4a to A₂>P and A₁>P, indicating asymmetric cleavage. Therefore, the structure of cA₃-bound Crn4a would be quite valuable to directly visualize how this asymmetric cleavage is achieved. If obtaining this structure is not feasible, can the authors provide a detailed discussion to address this point?

This is a good point. A good understanding of the relationship between cOA specificity and the structures of the two ring nucleases would require structures of each with cA₃ (and possibly cA₄) bound. Unfortunately, although we attempted to obtain such complexes, the data always showed no ligand was bound. This point is now discussed in the text:

In Results:

Despite our best efforts, we were unable to crystallise a ligand-bound complex with Crn4b, or indeed Crn4a in complex with cA₃ or cA₄. Given the considerable movement observed for Crn4a upon binding of cA₆ and our suggestion that specificity may largely be driven by interactions with the 'open' form of Crn4, direct comparison of the factors influencing specificity for substrate for each protein was not possible. Given the broad substrate specificity of the enzyme, this remains an important topic for future study.

In Discussion:

The observation that cA₃ is degraded efficiently by Crn4 is particularly intriguing, given that the enzyme has a dimeric organisation with two identical active sites and two-fold symmetry. While this is ideal for binding of 2-fold symmetric substrates such as cA₄ and cA₆, it poses a challenge for binding and cleavage of cA₃, which has 3-fold symmetry. An understanding of cA₃ binding and cleavage at the molecular level requires a co-crystal structure, but unfortunately we were unsuccessful in obtaining this structure. In its absence, we can speculate that cA₃ engages part of the binding site and one of the two catalytic sites, leading to asymmetric cleavage.

2. Page 6, 2nd paragraph: the authors drew a conclusion that Csm6-2 cleaves cOA using its CARF domain rather than HEPN domain, based on the result that EDTA did not inhibit cOA cleavage. However, as previously noted (Du et al. 2024, 43, 304), EDTA does not effectively inhibit HEPN domain RNase activity of TtCsm6. I recommend using a Csm6-2 mutant lacking HEPN domain activity to specifically address which domain is responsible for its intrinsic ring nuclease activity on cA₆.

Thanks for pointing this out. As Csm6-2 is not the focus of this paper we have opted to revise the text to make clear that the observed ring nuclease activity of Csm6-2 could be due to its CARF domain, HEPN domain, or a combination of the two.

3. Page 7, 1st paragraph: In the structural analysis of cA₆ binding, the authors identify several residues (Arg92, Glu16, Ser91, Arg112, etc.) involved in cA₆ interaction. However, these residues are not indicated in the figures. Please explicitly label these key residues in Fig. 4C and 4D to visually link the text description to the structural context. Also please show the labeling for residues in Fig. 4A-B

Residues are now labelled in the figures, as requested. Additional figures showing the interactions have been added as Supplementary Figures 9D and 9E.

4. To support the claim of specific cA₆ binding, please include a panel in Fig. 4 showing the 2F_o-F_c electron density map for the bound cA₆ ligand.

This has been included as Supplementary Figure 9A.

5. Please standardize to residue abbreviations (one-letter or three-letter) in all text, figure legends, and labels.

This has been done as requested, using the one letter codes.

Reviewer 1:

Did authors intend to use "surprisingly" instead of "unsurprisingly" in the sentence "Perhaps unsurprisingly, cA5, which is not known to be an activator of any CRISPR effector, was degraded efficiently by Crn4a." ?

Response: We meant unsurprisingly, given Crn4 cleaves all other cOA species. We have now clarified this in the text.

It is better to add the requirement for high-concentrations of polyA in the sentence "12 nt polyA RNA substrate was cleaved by Crn4a three nucleotides from the 3' end at an elevated concentration."

Response: We rephrased for clarity: "and a high concentration of Crn4 cleaved a 12 nt polyA RNA substrate three nucleotides from the 3' end."

Authors could note that "The fact that Crn4a, which lacks a HEPN domain, can cleave polyA suggests its substrate plasticity."

Response: We added a phrase on substrate plasticity to the discussion.

Figure S7. Panel A, can authors clarify what the cleavage products of cA3 and cA5 likely are? They do not seem to be aligned on the chromatogram.

Response: We re-ran these samples and have labelled them in figure S7